# Prevalence of Gestational Diabetes in Triplet Pregnancies: A Retrospective Cohort Study and Meta-Analysis

**DOI:** 10.3390/jcm9051523

**Published:** 2020-05-18

**Authors:** Marlene Hager, Johannes Ott, Deirdre Maria Castillo, Stephanie Springer, Rudolf Seemann, Sophie Pils

**Affiliations:** 1Clinical Division of Gynecologic Endocrinology and Reproductive Medicine, Department of Obstetrics and Gynecology, Medical University of Vienna, Spitalgasse 23, 1090 Vienna, Austria; marlene.hager@meduniwien.ac.at (M.H.); deirdre.castillo@meduniwien.ac.at (D.M.C.); 2Clinical Division of Obstetrics and Fetomaternal Medicine, Department of Obstetrics and Gynecology, Medical University of Vienna, Spitalgasse 23, 1090 Vienna, Austria; stephanie.springer@meduniwien.ac.at; 3Department of Oral and Maxillofacial Surgery, Medical University of Vienna, Spitalgasse 23, 1090 Vienna, Austria; rudolf.seemann@gmail.com; 4Clinical Division of General Gynecology and Gynecologic Oncology, Gynecologic Cancer Unit, Comprehensive Cancer Center, Medical University of Vienna, 1090 Vienna, Austria; sophie.pils@meduniwien.ac.at; 5Department of Obstetrics and Gynecology, Medical University of Vienna, Spitalgasse 23, 1090 Vienna, Austria

**Keywords:** triplets, multiple gestation, gestational diabetes, oral glucose tolerance test, pregnancy outcome

## Abstract

Background: Over the last decades, there has been a substantial increase in the incidence of higher-order multiple gestations. Twin pregnancies are associated with an increased risk of gestational diabetes mellitus (GDM). The literature on GDM rates in triplet pregnancies is scarce. Methods: A retrospective cohort study was performed to assess the prevalence of GDM in women with a triplet pregnancy. GDM was defined through an abnormal oral glucose tolerance test (OGTT). A meta-analysis of GDM prevalence was also carried out. Results: A cohort of 60 women was included in the analysis. Of these, 19 (31.7%) were diagnosed with GDM. There were no differences in pregnancy outcomes between women with and without GDM. In the meta-analysis of 12 studies, which used a sound GDM definition, an estimated pooled prevalence of 12.4% (95% confidence interval: 6.9–19.1%) was found. In a leave-one-out sensitivity analysis, the estimated GDM prevalence ranged from 10.7% to 14.1%. Conclusion: The rate of GDM seems increased in women with triplets compared to singleton pregnancies. However, GDM did not impact short-term pregnancy outcomes.

## 1. Introduction

Over the last decades, there has been a substantial increase in the incidence of higher-order multiple gestations because of assisted reproductive techniques [1]. Although there is a current downtrend due to restrictions in the numbers of transferred embryos and fetal reduction procedures [2,3], higher multiple pregnancies are clinically important because of their association with both increased maternal and fetal/neonatal morbidity and mortality [4].

During a healthy pregnancy, the mother undergoes several physiologic adaptions in order to provide the best support for the growing fetus. These include alterations in insulin sensitivity, i.e., an increase in early and a decrease in later pregnancy [5]. The latter is also caused by increases in local and placental hormones, including estrogen, progesterone, leptin, cortisol, placental lactogen, and placental growth hormone [6]. It has been claimed that this “diabetogenic effect” of pregnancy may be accentuated in multiple gestations due to increased placental mass, and thus, increased placental hormones [7,8,9]. Moreover, assisted reproductive techniques (ART) are associated with an increase in maternal age compared to spontaneously conceived pregnancies, and increased maternal age is in turn independently associated with increased gestational diabetes mellitus (GDM) prevalence.

Higher GDM risks have been reported for twin pregnancies [10]. However, only limited data are available in triplet pregnancies [7,11,12,13,14,15], likely due to their rarity. Thus, further data are needed Our aim was to present a retrospective analysis of GDM incidence in triplet pregnancies managed at our department. In addition, we included our data into a meta-analysis of available reports about this topic.

## 2. Materials and Methods

### 2.1. Patient Population and Study Design

A total of 138 women who were diagnosed with a triplet pregnancy at the time of the first trimester screening from January 2003 to April 2018 were included in our analysis. The following patients were excluded from the study: all women who underwent multifetal pregnancy reduction (*n* = 15) or had an intrauterine fetal death (IUFD) of at least one fetus before the onset of viability (*n* = 6); women who did not deliver at the department, and thus, were lost to follow-up (*n* = 7); women with a late miscarriage or a delivery <24th completed week of gestation (*n* = 16). Routinely, an oral glucose tolerance test (oGTT) is performed between the 24th and the 28th completed gestational week in Austria, and accordingly, no oGTT results were available for these women; and women who did not undergo an oGTT due to delivery prior to the appointment (*n* = 22) or because they refrained from it against all recommendations (*n* = 6). This resulted in a final study population of 60 triplet pregnancies with 180 fetuses/neonates who were included in this analysis. Parts of these data have been published previously with a focus on cervical length measurements [16] and fetal weight estimation [17]. These 60 women with triplet pregnancies were compared to 60 matched women with singleton pregnancies (1:1 matching for age and body mass index (BMI)). The latter were selected from the large population of pregnant women who had undergone first trimester screening from January 2003 to April 2018 and had subsequently delivered at the department.

The Department of Maternal–Fetal Medicine of the Medical University of Vienna, Austria, is the reference center for maternal–fetal medicine in eastern Austria with an annual number of deliveries of ≥2500 during the study period. As reported previously [17], all women had undergone sonographic screening, which included fetal biometry using Hadlock’s formula [18] and a cervical length measurement by transvaginal ultrasound every two weeks from week 16 + 0 until delivery. All ultrasound examinations were performed by highly experienced obstetricians, all members of the clinical working group for multiple pregnancies, and were performed on the same two ultrasound devices: a Toshiba Power Vision (Toshiba, Tokyo, Japan) ultrasound machine was used until 2010, and a Toshiba Aplio MX (Toshiba, Tokyo, Japan) machine since 2010.

The basic perinatology database at the department uses the Viewpoint^®®^ software (GE Healthcare, Wessling, Germany), which was also used for data acquisition. The study was approved by the Institutional Review Board of the Medical University of Vienna (IRB number: 1523/2018) on 3 September 2018, and was valid for one year after approval. The study protocol was in accordance with the Helsinki Declaration and current Austrian law, and thus, neither written nor verbal informed consent was necessary according to the Ethics Committee of the Medical University of Vienna. Therefore, it was not obtained. The data were deidentified for statistical analysis.

### 2.2. Parameters Analyzed

GDM, the main outcome parameter, was assessed using a 75 g 2-h oGTT, according to the International Association of Diabetes in Pregnancy Study Group’s (IADPSG) recommendations [19]. It was performed from gestational weeks 24 + 0 to 28 + 0 and was rated as abnormal if one value exceeded the threshold (fasting: 92 mg/dL, 1 h: 180 mg/dL, 2 h: 153 mg/dL). In addition, the following parameters were obtained: gestational age at delivery (in completed weeks); maternal age at delivery (years); prepregnancy body mass index (BMI, kg/m^2^); parity; pregnancies after in vitro fertilization (IVF) or ovarian stimulation (i.e., clomiphene citrate, letrozole, or recombinant follicle stimulating hormone without IVF); cigarette smoking; pregnancy induced/preexisting hypertension; birthweight (g and in percentile, according to the data of Voigt et al. [20]); and chorionicity categorized into mono-/dichorionic and trichorionic for the multivariate analysis. All patients were delivered by Caesarean section.

### 2.3. Standard GDM Management

Universal testing by a 75 g OGTT at 24–28 weeks of gestation was used to diagnose GDM which is in accordance to the IADPSG recommendations [19,21]. Women were seen by both obstetricians and diabetologists. The first line intervention was intensified lifestyle modification including medical nutrition therapy. All patients were instructed for capillary blood glucose monitoring and informed about glycemic treatment targets. Follow-ups for two weeks later were scheduled and blood glucose levels were reviewed during each appointment. When blood glucose targets were not achieved (i.e., <95 mg/dL at fasting or <140 mg/dL one hour after each meal), pharmacologic intervention with insulin was started at any time point [22].

### 2.4. Meta-Analysis

For the systematic literature review, we searched the Medline database (search date: 1 July 2019; search terms: “((((triplet*) AND pregnancy)) NOT case report) NOT review)” to identify original cohort studies about triplet pregnancies. Three authors assessed the eligibility of the studies, extracted data on GDM prevalence, and assessed the risk of bias (M.H., D.M.C., and J.O.). Missing information and additional trials were not sought by the authors. A qualitative assessment of the studies included in the meta-analysis of studies with a sound GDM definition was also performed. Although not all items of the Newcastle–Ottawa Scale for cohort studies [23] were applicable to the meta-analysis, the items were used as far as possible. Concerning “selection”, we assessed whether the cohort was truly representative of the average triplet pregnancy. For the “comparability” of studies, we assessed whether age and BMI had been reported. For “outcome”, we evaluated whether the source for the retrospective dataset had been specified. Studies fulfilling all criteria were rated as having the lowest risk of bias, studies fulfilling two of the three items (“selection”, “comparability”, and “outcome”) were assessed as having a medium risk of bias, and studies fulfilling one or no criterion were considered to have the highest risk of bias. A qualitative assessment of studies was performed by two researchers (M.H., J.O.).

### 2.5. Statistical Analyses

Variables are described by numbers (frequencies) and median (interquartile ranges). The statistical analysis was performed with SPSS 25.0 for Windows (SPSS Inc., 1989–2019, IBM, Armonk, NY, USA) using the Fisher’s exact test for categorical parameters. Multivariate binary logistic regression models were used to test the predictive value of all coefficients for binary outcome parameters. Odds ratios (OR) and their 95% confidence intervals (95% CI) are given. Differences were considered statistically significant if *p* < 0.05.

The meta-analysis on the incidence of GDM in women with triplet pregnancies was performed as published previously [24]: The library “metafor” in the open source statistical package “R” (The R Project for Statistical Computing) was used. The observed proportions were transformed using the Freeman–Tukey double arcsine transformation, which provides an effect measure with a favorable sampling distribution and stable variance. A meta-analysis model was fit to the transformed data using inverse variance weights and including a random effect to account for between-study heterogeneity. The random effects model was used because of the differences in observed GDM prevalence in the included studies, in order not to underestimate the variability of data. A pooled estimate of the prevalence of GDM and the according 95% confidence interval (95% CI) were obtained by back-transforming the respective quantities to the original scale. In addition, the I^2^ value is provided, describing the percentage of variation across studies that is due to heterogeneity rather than chance.

## 3. Results

### 3.1. Retrospective Cohort Study

The median patients’ age was 32 years (Interquartile Range (IQR) 27–35); prepregnancy BMI was 23.8 kg/m^2^ (IQR 22.3–26.3). Nineteen women (31.7%) were diagnosed with an abnormal oGTT. This prevalence exceeded that in age- and BMI-matched singleton pregnancies (11.7%) significantly (*p* = 0.010). Details on the comparison between these two groups are provided in Table 1. As demonstrated in Table 2, there were no differences in basic patient characteristics and pregnancy outcomes between women with and without GDM apart from higher fasting, 1-h and 2-h plasma glucose levels during the 75 g oGTT in the GDM group. None of the patients affected with GDM became insulin dependent.

### 3.2. Meta-Analysis

A total of 2021 articles were identified. Of these, 1983 reports were excluded step-by-step. Details are provided in Figure 1. Thus, 38 studies were included in the initial meta-analysis in addition to the present case series. Furthermore, a second meta-analysis was performed which included only those studies with a sound definition of GDM (*n* = 11): for this meta-analysis, GDM had to be diagnosed using either the 3-h, the 100 g oGTT (*n* = 7), the 2-h 75 g oGTT (*n* = 3), or the 1-h glucose challenge test which—in case of any abnormality—was followed by a 3-h, 100 g oGTT (*n* = 1).

All eligible studies were included in the pooled models. After correction for study heterogeneity, the estimated GDM prevalence was 7.3% (95% CI: 5.0–9.9) for the analysis of all 38 eligible studies [9,25,26,27,28,29,30,31,32,33,34,35,36,37,38,39,40,41,42,43,44,45,46,47,48,49,50,51,52,53,54,55,56,57,58,59,60] (Figure 2) with an I^2^ of 93.2% for total study heterogeneity. The estimated GDM prevalence was 12.4% (95% CI: 6.9–19.1), when only studies with a sound GDM definition were used [9,33,37,41,43,48,52,53,55,59] with an I^2^ of 83.2% (Figure 3). Table 3 provides an overview on the GDM definitions used and other study characteristics in the second meta-analysis. The lowest, medium, and highest risk of bias was found for two studies [53] (and for our own dataset), six studies [9,33,40,41,43,48], and three studies [37,52,55], respectively (Table 3). The respective funnel plots are shown in Figure 4. All studies were plotted near the average. In the leave-one-out sensitivity analyses, the estimated GDM prevalence ranged from 6.7% to 7.7% (Appendix A) and from 10.7% to 14.1% (Appendix A) for the complete and the partial models, respectively.

## 4. Discussion

According to the two meta-analyses, 7.3–12.4% of women with triplet pregnancies suffered from GDM. The difference in GDM rates between the two analyses seems clinically relevant. It is likely that this is due to the fact that in the meta-analysis of all 38 studies, which somehow evaluated the GDM prevalence in triplet pregnancies (Figure 1), many did not provide the exact diagnostic criteria for GDM or used definitions that are not considered standard nowadays. We, thus, consider the second meta-analysis (*n* = 11; Figure 2) with a standardized and currently valid GDM definition (Table 2) more reliable. Accordingly, a pooled GDM prevalence of 12.4% was found in triplet pregnancies. It has been demonstrated that twin pregnancies carry a higher risk of GDM development than singleton pregnancies [8]. Notably, a wide range of GDM prevalence, from about 7% to 20%, has been reported for twin pregnancies [8,61,62,63]. It seems obvious that the results of our meta-analyses lie within that range. Thus, one cannot state that women with triplets would carry a higher GDM risk than those with twins. However, women with triplets are obviously at an increased risk of GDM compared to women with singleton pregnancies, a fact that was also demonstrated by the matched comparison in our patient population (Table 1).

When focusing on the studies included into our meta-analyses, even the studies with sound GDM definitions, a clinically relevant range of the GDM prevalence (7.4–38.5%) becomes evident (Figure 2) which is also reflected by the considerably high I^2^ of about 83%. We find it hard to comment on this finding, although these differences might be due to the different GDM definitions used (Table 2), which we consider less likely, or to differences in the study populations. Literature directly comparing the different tools for GDM diagnosis is scarce. One prospective study demonstrated a higher GDM rate with the 75 g oGTT than with the 100 g oGTT and raised the suspicion that the latter would be more specific for detecting GDM complications [64]. However, as can be seen in Table 3 and Figure 2, this was not the case for the studies included in the meta-analysis. However, one has to be aware that nine out of eleven studies showed a medium to high risk of bias which must be considered a limitation of the meta-analysis. Nonetheless, as shown in the funnel plot (Figure 3) and the leave-one-out sensitivity analyses (Appendix A), the data of the meta-analysis only showed a moderate distribution.

As discussed above, multiple pregnancies obviously carry a higher GDM risk. From a pathophysiological point of view, pregnancy itself has been blamed for exerting “diabetogenic effects”. Especially in later pregnancy, insulin sensitivity usually declines [5], which is believed to be caused by increases in several endocrine and paracrine factors. The latter include estrogen, progesterone, leptin, cortisol, placental lactogen, and placental growth hormone [6]. It has been hypothesized that placenta-derived hormones might be positively correlated with placenta mass, and thus, women with multiple pregnancies would be burdened by a higher GDM risk [7,8,9]. In addition, the use of artificial reproductive techniques is associated with the development of multiple gestations, as well as with a higher age of treated patients. This could also contribute to the increased GDM prevalence in women with multiple pregnancies.

Notably, when focusing on our retrospective dataset, the median maternal age was 32 years, and the median BMI was about 23.5 kg/m^2^ (Table 1). Moreover, there were no differences between the GDM and the non-GDM groups concerning age and BMI. Thus, we believe that these did not contribute to GDM development. However, the high rate of women who smoked during their triplet pregnancy is to be noted. This is a particular feature which became evident in the course of previous studies in our department [16,65]. This seems to be of clinical relevance, since prenatal smoking has been associated with higher odds of GDM according to a recent analysis [66]. Despite this association not reaching statistical significance in our dataset (Table 1), a trend can be seen (OR 4.524, *p* = 0.057). Hypothetically, this might have contributed to the quite high GDM prevalence in our dataset (31.7%).

What can also be seen from this analysis is that GDM might not have a high impact on short-term outcome of triplet pregnancies. As demonstrated in Table 1, there were no differences in gestational age at delivery between women with and without GDM. One can assume that this was due to the fact that triplets usually were delivered preterm. Moreover, GDM was not associated with differences in birthweight. Notably, fetal growth can be compromised by various mechanisms in triplet pregnancies [67]. Moreover, the women were being managed and controlled according to our department’s standard procedures (see the Methods Section). Thus, the lack of differences in BMI between the GDM- and non-GDM patients could also be seen as a reflection of good care being provided for these women. They were also being delivered at a median gestational age of 33 completed weeks. At this particular time, fetal fat deposition is presumably not at its maximum and a subtle impact of birth weight might not yet be evident. However, we do believe that screening for GDM in triplet pregnancies is still of clinical relevance, since women with GDM carry a risk of progression to type 2 diabetes [68]. The fact that short-term outcome was not affected by GDM might also be due to the fact that none of the patients became insulin dependent.

One has to address several limitations of this report: first, the majority of studies included in the meta-analyses were of retrospective design (Table 2); secondly, they used different definitions of GDM, even in the second smaller analysis (Table 2); thirdly, this also holds true for our dataset which only included a small sample size. This might limit the claim that GDM was not associated with short-term obstetric outcome in triplets. Last but not least, insulin-dependent GDM could not be distinguished from non-insulin-dependent GDM in the meta-analyses.

## 5. Conclusions

Finally, it can be concluded that the rate of GDM seems slightly increased in women with triplets compared to singleton pregnancies. We consider the latter finding sound, since GDM prevalence was evaluated in a well-characterized cohort of women cared for in a tertiary referral center providing care in line with internationally accepted standards of practice. However, the prevalence was within the range reported for twin pregnancies [8,61,62,63]. Notably, GDM did not impact short-term pregnancy outcome in the retrospective dataset. Whether women with triplets should undergo early screening, and whether measures should be taken to reduce the associated risks remains open for future research.

## Figures and Tables

**Figure 1 jcm-09-01523-f001:**
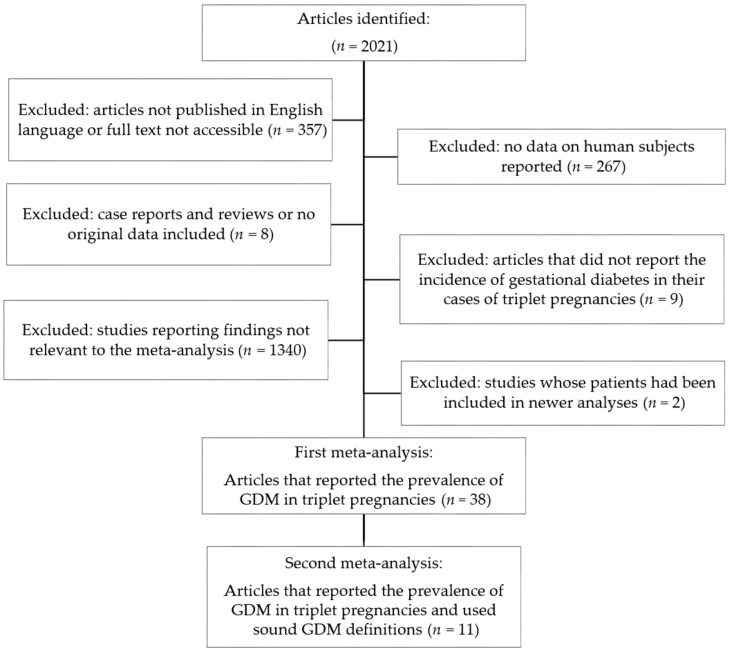
Flow chart for the meta-analysis. GDM, gestational diabetes mellitus.

**Figure 2 jcm-09-01523-f002:**
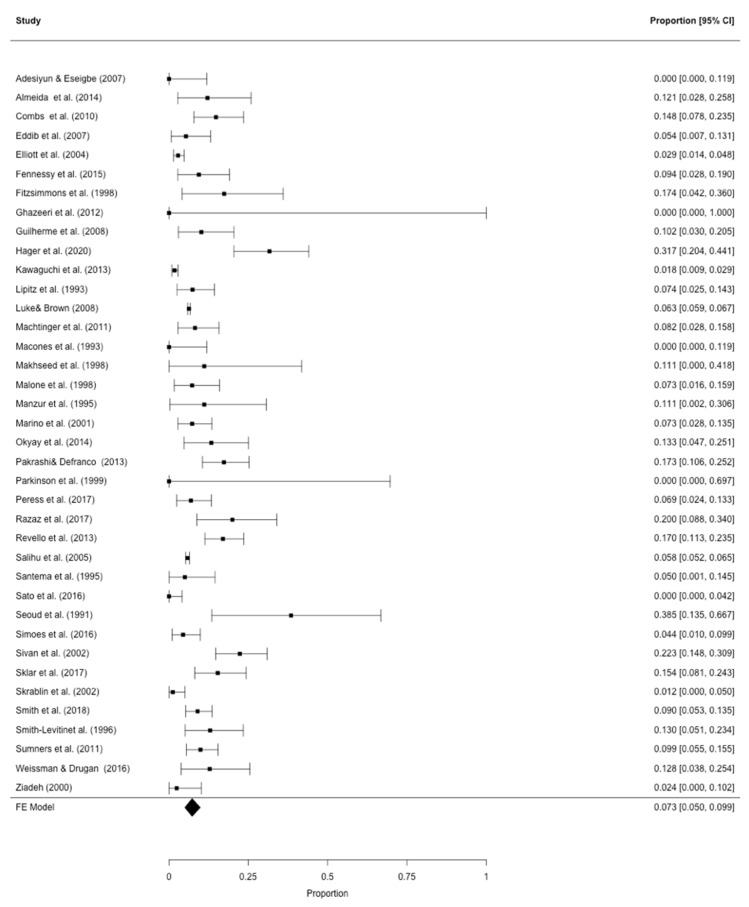
Meta-analysis of all 38 eligible studies on GDM prevalence in triplets [9,25,26,27,28,29,30,31,32,33,34,35,36,37,38,39,40,41,42,43,44,45,46,47,48,49,50,51,52,53,54,55,56,57,58,59,60].

**Figure 3 jcm-09-01523-f003:**
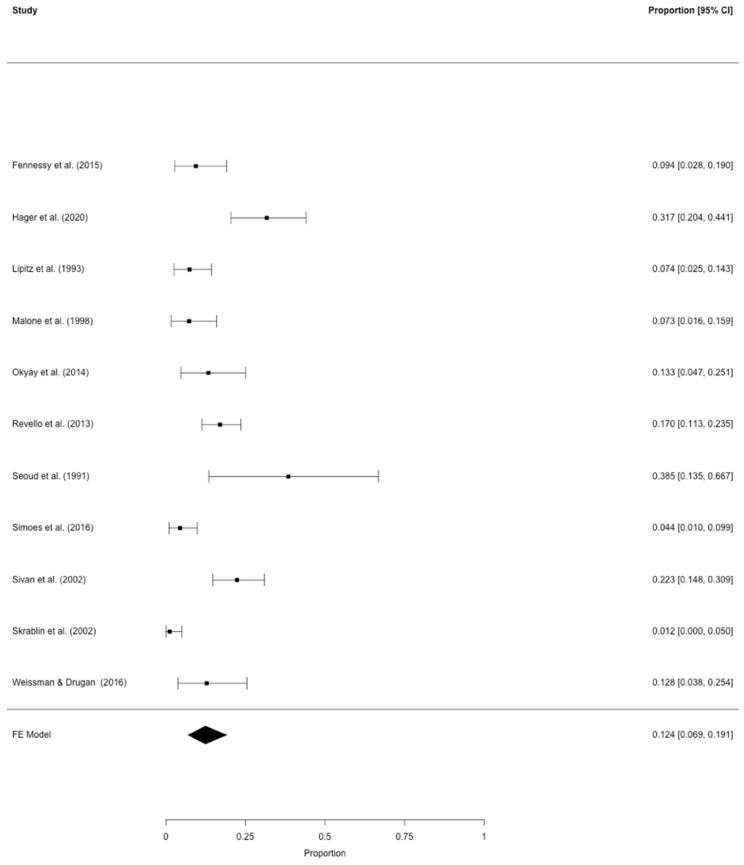
Meta-analysis of all 11 studies on GDM prevalence in triplets that used a sound GDM definition [9,33,37,41,43,48,52,53,55,59].

**Figure 4 jcm-09-01523-f004:**
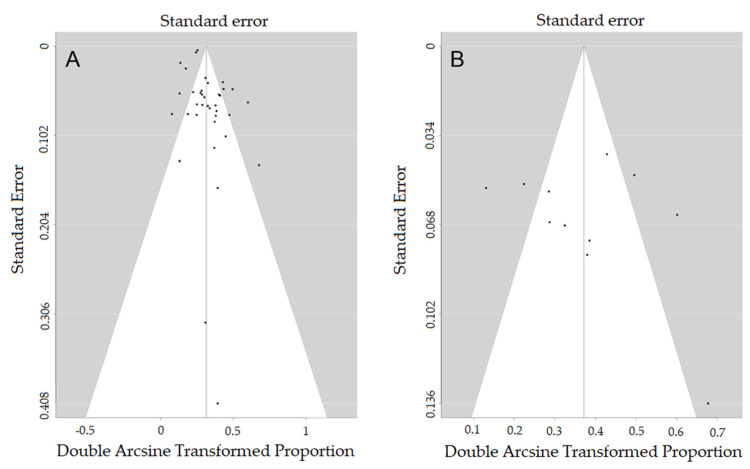
Funnel plots for (**A**) the meta-analysis of all eligible studies and (**B**) the meta-analysis of all studies with a sound GDM definition.

**Table 1 jcm-09-01523-t001:** Triplet compared to age- and BMI-matched singleton pregnancies: basic patient characteristics and outcomes concerning gestational diabetes mellitus (GDM).

		Triplet Pregnancies (*n* = 60)	Singleton Pregnancies (*n* = 60)	OR (95% CI)	*p* (LR Test)
Age (years)		32 (27;35)	32 (27;35)	1.000 (0.934;1.071)	1.000
BMI (kg/m^2^)		23.8 (22.3;26.3)	23.7 (22.4;26.5)	0.999 (0.904;1.104)	0.984
Parity	0	44 (73.3)	25 (41.7)		0.002
	1	12 (20.0)	30 (50.0)	0.227 (0.099;0.521)	
	≥2	4 (6.7)	5 (8.3)	0.455 (0.112;1.850)	
Pregnancy after IVF or ovarian stimulation		43 (71.7)	7 (11.7)	19.151 (7.276;50.407)	*<0.001*
Smoking during pregnancy		8 (13.3)	11 (18.3)	0.685 (0.254;1.846)	0.455
GDM (abnormal oGTT)		19 (31.7)	7 (11.7)	3.509 (1.347;9.142)	*0.010*
75 g oGTT: fasting glucose level (mmol/L)		4.66 (4.38;5.05)	4.55 (3.94;4.83)	1.136 (1.044;1.237)	*0.003*
75 g oGTT: 1-h glucose level (mmol/L)		8.71 (7.44;9.82)	6.87 (5.61;8.67)	1.075 (1.034;1.118)	*<0.001*
75 g oGTT: 2-h glucose level (mmol/L)		6.67 (5.61;8.05)	5.49 (4.32;6.67)	1.049 (1.022;1.077)	*<0.001*

Significant *p*-values are provided in italics; Abbreviations used: BMI, body mass index; oGTT, oral glucose tolerance test; GDM, gestational diabetes mellitus; IVF, in vitro fertilization; OR, odds ratios; LR, likelihood ratio.

**Table 2 jcm-09-01523-t002:** Triplet pregnancies with and without GDM, defined as an abnormal 75 g, 2-h oGTT.

		Abnormal oGTT (*n* = 19)	Normal oGTT (*n* = 41)	OR (95% CI)	*p* (LR test)
Age (years)		32 (28;36)	32 (27;35)	0.988 (0.900;1.086)	0.805
BMI (kg/m^2^)		23.5 (22.7;27.3)	23.7 (22.3;26.1)	1.033 (0.885;1.206)	0.677
Parity	0	15 (78.9)	29 (70.7)		
	1	4 (21.1)	8 (19.51)		
	2	0	4		
Pregnancy after IVF		13 (68.4)	25 (61.0)	1.950 (0.536;7.088)	0.306
Pregnancy after ovarian stimulation		1 (3.1)	4 (10.0)	0.563 (0.058;5.440)	0.615
Smoking during pregnancy		5 (23.8)	3 (7.3)	4.524 (0.953;21.464)	0.057
Pregnancy induced/preexisting hypertension		1 (5.3)	4 (9.8)		0.558
Gestational age at delivery (completed weeks)		33.0 (31.7;34.0)	33.1 (31.4;34.0)	1.014 (0.978;1.051)	0.457
Median birthweight (g)		1691 (1517;1893)	1673 (1299;1933)	1.001 (0.999;1.002)	0.43
Median birth weight (percentile)		33.8 (14.4;62.2)	43.1 (24.5;69.2)	0.990 (0.978;1.002)	0.098
75 g oGTT: fasting glucose level (mmol/L)		5.16 (4.66;5.66)	4.55 (4.27;4.72)	1.136 (1.044;1.237)	*0.003*
75 g oGTT: 1-h glucose level (mmol/L)		10.55 (9.66;11.60)	7.88 (7.10;8.82)	1.075 (1.034;1.118)	*<0.001*
75 g oGTT: 2-h glucose level (mmol/L)		8.56 (6.55;10.27)	6.16 (5.44;7.16)	1.049 (1.022;1.077)	*<0.001*

Significant *p*-values are provided in italics; Abbreviations used: BMI, body mass index; oGTT, oral glucose tolerance test.

**Table 3 jcm-09-01523-t003:** Meta-analysis of studies with a sound GDM definition: overview on study design and GDM definitions used.

First Author (Year)	Study Design	Number of Subjects	Country	GDM Definition	Selection of Cases	Mean/Median Prepregnancy BMI (kg/m^2^)	Mean/Median Maternal Age (Years)	Source of Data Specified	Risk of Bias
Fennessy (2015) [33]	Retrospective	53	Australia	75 g, 2-h oGTT	Representative	Not reported	31.6–32.9	Yes	Medium
Hager (2020)	Retrospective	60	Austria	75 g, 2-h oGTT	Representative	23.5	32.0	Yes	Lowest
Lipitz (1993) [37]	Retrospective	81	Israel	100 g, 3-h oGTT	Representative	Not reported	31.2	No	Highest
Malone (1998) [41]	Retrospective	55	USA	100 g, 3-h oGTT	Representative	Not reported	32.0	Yes	Medium
Okyay (2014) [43]	Retrospective	45	Turkey	50 g, 1-h oGTT, followed by a 100 g, 3-h oGTT	Representative	23.6–26.0	28.9–30.1	No	Medium
Revello (2013) [48]	Retrospective	147	Spain	100 g, 3-h oGTT	Representative	Not reported	34.0	Yes	Medium
Seoud (1991) [52]	Retrospective	13	USA	100 g, 3-h oGTT	IVF only	Not reported	32.2	Yes	Highest
Simoes (2016) [53]	Retrospective	90	Israel	100 g, 3-h oGTT	Representative	23.6–23.9	32.1–33.1	Yes	Lowest
Sivan (2002) [9]	Retrospective	103	Israel	1-h glucose challenge test, followed by a 100 g, 3-h oGTT	Representative	23.9	29.2	No	Medium
Skrablin (2002) [55]	Retrospective	e85	Croatia	75 g, 2-h oGTT	Representative	Not reported	Not reported	No	Highest
Weissman (2016) [59]	Retrospective	39	Israel	100 g, 3-h oGTT	Representative	Not reported	30.9	Yes	Medium

## Data Availability

The datasets generated and/or analyzed during the current study are not publicly available, since the dataset will be used for other retrospective analyses. The data are available from the corresponding author upon reasonable request.

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
