# Peer review of "Prevalence of Gestational Diabetes in Triplet Pregnancies: A Retrospective Cohort Study and Meta-Analysis"

_jcm, 2020, doi:10.3390/jcm9051523_

Round 1

Reviewer 1 Report

Article on an interesting subject. The presentation of the results and the method must be improved.

Minor comments :

  • Specify the meaning of oGTT once
  • glucose values should be given in mmol/L

Major comments :

  • In first part of their study, the authors collected birth weight (line 94). This parameter should be analyzed in Table 1 (in percentile).
  • The PRISMA checktlist is missing. The PRISMA 27 checklist items pertain to the content of a systematic review and meta-analysis, which include the title, abstract, methods, results, discussion and funding.
  • A flow chart about meta-analysis is missing in a first figure.
  • For the 2 forest plots, the authors should give I² value to evaluate the heterogeneity of studies and justify the funnel plots.
  • In table 2, a lot of characteristics of studies included in the meta-analysis are missing: number of subjects, pre-pregnancy-BMI, ethnicity, age of mothers, quality assessment and risk of bias in the reviewed studies.

Author Response

Dear Reviewer,

we thank you for the time and effort you put in the review-process of our manuscript. We agree with your recommendations and took care in revising our manuscript accordingly. Please find a point-by-point answer letter below. We hope that the changes made, will make our manuscript acceptable for publication in the “Journal of Clinical Medicine”. If any queries should remain, we should be happy to address them in the course of future revisions.

Respectfully yours,

Johannes Ott

Reviewer number 1

Minor comments:

Specify the meaning of oGTT once

Reply: Corrected as recommended.

glucose values should be given in mmol/L

Reply: Corrected as recommended. Please see revised Table 1.

Major comments:

In first part of their study, the authors collected birth weight (line 94). This parameter should be analyzed in Table 1 (in percentile).

Reply: We thank the reviewer for this input. The following data were added to Table 1:

Median birth weight (percentile): GDM: 33.8 (14.4;62.2); no GDM: 43.1 (24.5;69.2); odds ratio with 95% CI: 0.990 (0.978;1.002); p= 0.098

In addition, we revised the Methods Section: “In addition, the following parameters were obtained: […] birth weight (g and in percentile, according to the data of Voigt et al. [19]); […]”

New reference: 19. Voigt M, Olbertz D, Hentschel R, Kunze M, Hagenah HP, Scholz R, Wittwer-Backofen U, Hesse V, Straube S (2016) Perzentilwerte für die Körpermaße neugeborener Drillinge – Ergebnisse der deutschen Perinatalerhebung der Jahre 2007–2011 unter Beteiligung aller Bundesländer. Z Geburtshilfe Neonatol 220(2):66-73.

The PRISMA checklist is missing. The PRISMA 27 checklist items pertain to the content of a systematic review and meta-analysis, which include the title, abstract, methods, results, discussion and funding.

Reply: We agree and added the completed PRISMA checklist as a supplementary material.

A flow chart about meta-analysis is missing in a first figure.

Reply: A flow chart about the in- and exclusion process of studies is presented in the new Figure 1.

For the 2 forest plots, the authors should give I² value to evaluate the heterogeneity of studies and justify the funnel plots.

Reply: We added the following sentences to the manuscript:

Methods (Statistical Analysis): “In addition, the I2 value is provided which describes the percentage of variation across studies that is due to heterogeneity rather than chance.”

Results (Meta-analysis): “After correction for study heterogeneity the estimated GDM prevalence was 7.3% (95% CI: 5.0-9.9) for the analysis of all 38 eligible studies [9,24-59] (Figure 2) with an I2 of 93.2% for total study heterogeneity. The estimated GDM prevalence was 12.4% (95% CI: 6.9-19.1), when only studies with a sound GDM definition were used [9,32,36,40,42,47,51,52,54,58] with an I2 of 83.2% (Figure 3).”

Discussion: “When focusing on the studies included into our meta-analyses, even the studies with sound GDM definitions, a clinically relevant range of the GDM prevalence (7.4% – 38.5%) becomes evident (Figure 2) which is also reflected by the considerably high I2 of about 83%.”

In table 2, a lot of characteristics of studies included in the meta-analysis are missing: number of subjects, pre-pregnancy-BMI, ethnicity, age of mothers, quality assessment and risk of bias in the reviewed studies.

Reply: We thank the reviewer for this important comment. We added the above mentioned characteristics to Table 2 (i.e. Table 3 in the revised manuscript).

- Methods: “Qualitative assessment of the studies included in the meta-analysis of studies with a sound GDM definition was also performed. Although not all items of the Newcastle-Ottawa Scale for cohort studies [ref] were applicable to the meta-analysis, the items were used as far as possible. Concerning “selection”, we assessed whether the cohort was truly representative of the average triplet pregnancy. For the “comparability” of studies, we assessed whether age and BMI had been reported. For “outcome”, we evaluated whether the source for the retrospective data set had been specified. Studies fulfilling all criteria for were rated as having lowest risk for bias, studies fulfilling two of three items (“selection”, “comparability”, and “outcome”) were assessed as having a medium risk for bias and studies fulfilling one or no criterion were considered to have the highest risk for bias. Qualitative assessment of studies was performed by two researchers (M.H., J.O.).”

- Results: “The lowest, medium, and highest risk for bias was found for two, six, and three studies, respectively (Table 3).”

- Revised Table 3.

- Discussion: “However, one has to be aware that nine/11 studies showed a medium to high risk for bias which must be considered a limitation of the meta-analysis.”

Reviewer 2 Report

This study aims to describe the prevalence of GDM in women with a triplet pregnancy and to compare this to previously reported GDM prevalence rates from other studies. 

Overall this is a worthwhile study since, as the authors point out, robust data on GDM prevalence in triplet pregnancies are lacking.  There are 60 pregnancies included in this analysis which make the findings worthy of publication.

However I think there are changes required to improve the quality of the publication. 

1) Editing required: Methods are described in results section and results are described in methods section in both the abstract the article

Abstract: 

Line 27 onwards (suggestion) -Methods: A retrospective cohort study was performed to assess the prevalence of GDM in women with a triplet pregnancy.  GDM was defined as an abnormal oral glucose tolerance test (OGTT). A meta-analysis of GDM prevalence was also performed. 

Results: A cohort of 60 women was included in the analysis.  Of these, 19 (31.7%) were diagnosed with GDM.... 

Materials and Methods/Results: 

The numbers studied should be included in the results section as well as the numbers excluded for various reasons. This should not be part of the methods section. 

2) The conclusion is not supported by the results.  Firstly, the conclusion states that GDM prevalence is increased in triplet pregnancies but it doesn't define in relation to what is it increased.  Is it increased in relation to singleton pregnancies or to twin pregnancies? A baseline prevalence then needs to be given for GDM prevalence in the comparison group (singletons or twins).  Secondly, the authors state that 'GDM seems of only minor impact' and yet this minor impact is not described as the outcomes are the same in both GDM and non-GDM groups.  Would it be better for the conclusion to state to state that 'GDM did not impact pregnancy outcome'?

3) I would suggest removing the reference to fetal weight in line 57.  Do the authors mean birth weight or estimated fetal weight on ultrasound? It is obvious that any study of GDM outcomes should include birthweight.  I would disagree that they 'focussed on this issue' as it is not analysed more than any other of the secondary outcomes reported. Throughout the paper the authors use the terms 'birthweight', 'neonatal weight' and 'fetal weight' all to apparently mean birthweight.  It would be best to use one term consistently. 

4) Lines 94 and 97 describe baseline characteristics obtained and describe both birthweight (line 94) and neonatal weight (line 97) what is the difference between these?  This whole section (line 93-100) needs revision as the units used are quoted for some parameters and not others.  This needs to be consistent.

5) Lines 99-100.  This statement seems strange as clinical examination does not rule out all malformations.  Do the authors mean that the babies were clinically well? What is the relevance of this to their study?  They have not reported other neonatal well-being measures such as Apgar scores, need for resuscitation, cord gas measurements etc.

6) It would be helpful to provide a brief description of management of GDM in the Hospital.  Are the women seen by both Obstetricians and Diabetologists?  How intensively are capillary glucose readings measured?  What are the initial interventions?  Is it dietary modification and exercise? What is the threshold for initiation of medication? Is Metformin used or is Insulin the first-line medical treatment?

6) Lines 129-32.  Why is there a reference to endometriosis?  Were there no other variations in the studies except endometriosis prevalence.  What relevance does this have to GDM prevalence?  This needs to be explained as it is confusing in its current format since there is no prior mention of endometriosis at any stage in the paper.

7) Of the 60 pregnancies 14 of the GDM group and 29 of the non-GDM group were the results of IVF or ovarian stimulation.  This implies that the other pregnancies were spontaneously conceived. The means that 16/60 triplet pregnancies were spontaneously conceived.  This seems to be an abnormally high proportion.  Is this correct?

8) Table 1: 'Birth weight' is more correct than 'neonatal weight'

9) Line 169 - 'notably a wide range of GDM prevalence.. has been reported'.  Is this for twin or triplet pregnancies? 

10) Lines 189-192 - the authors are again commenting on the association between ART and maternal age as they have previously done in lines 50-2.  However the maternal age was not different in their GDM and non-GDM groups.  Also the mean maternal age in both groups was only 32 years which is relatively young compared to other reports from ART cohorts.  The authors should be decisive as to whether or not they feel that maternal age was contributory to GDM prevalence in their cohort.  It would be helpful if they compared their triplet cohort to the non-triplet cohort in their hospital in terms of the baseline parameters reported in Table 1. 

11) Lines 206-8.  Do the authors really think that the lack of difference in bithweight between the two groups is due to the fact that the GDM women didn't require insulin?  This does not seem to be a careful consideration of the factors at work.  These are triplet pregnancies with multiple reasons for compromise of fetal growth.  Similarly (I assume) these women are being managed and controlled well (see point 6 above).  I would see this lack of differences as a reflection on good care being provided for these women.  They are also being delivered at 33 weeks gestation, at a time when fetal fat deposition is not at its maximum and a subtle impact of birth weight might not yet be evident.  The Discussion could be revised and strengthened by acknowledging these factors. 

12) The conclusion needs to be revised.  I think this study is important because it presents a value for GDM prevalence in triplet pregnancies in a well-characterised cohort of women cared for in a tertiary referral centre providing care in line with internationally accepted standards of practice.  I think this is the most important point of the study.  As the authors acknowledge, there are limitations with the data included in the meta-analysis, therefore this is of minor scientific interest.  It is necessary to compare the GDM prevalence in triplets to that of either singletons or twins in their centre.  The most important finding in this study is that the prevalence of GDM in triplet pregnancies is 32%.  

13) Language: It is obvious that this paper is not written by authors who use English as a first language and there are many sentences that read awkwardly.  I would recommend review by an English language editor prior to publication.  Examples of some areas where language could be improved include: 

Abstract: 

Line 24 - 'there has been an substantial increase..'

Line 25 - 'twin pregnancies are associated with...'

Line 26 - 'gestational diabetes mellitus (GDM)'

Lines 29 and 136 - diagnosed rather than revealed

Introduction: 

Line 40 - 'gestations because of assisted reproductive techniques..'

Lines 42-3 (suggestion) - are clinically important because of their association with both increased maternal and fetal/neonatal morbidity and mortality

Lines 50-2 (suggestion) - Assisted reproductive techniques (ART) are associated with an increase in maternal age compared to spontaneously conceived pregnancies and increased maternal age is in turn independently associated with increased GDM prevalence

Lines 54-5 - 'further data are needed'

Lines 170-4 - these sentences needs to be re-phrased.  It is not clear what the authors are trying to say. 

Author Response

Dear Reviewer,

we thank you for the time and effort you put in the review-process of our manuscript. We agree with your recommendations and took care in revising our manuscript accordingly. Please find a point-by-point answer letter below. We hope that the changes made, will make our manuscript acceptable for publication in the “Journal of Clinical Medicine”. If any queries should remain, we should be happy to address them in the course of future revisions.

Respectfully yours,

Johannes Ott

Reviewer number 2

1) Editing required: Methods are described in results section and results are described in methods section in both the abstract the article

Abstract:

Line 27 onwards (suggestion) -Methods: A retrospective cohort study was performed to assess the prevalence of GDM in women with a triplet pregnancy. GDM was defined as an abnormal oral glucose tolerance test (OGTT). A meta-analysis of GDM prevalence was also performed.

Results: A cohort of 60 women was included in the analysis. Of these, 19 (31.7%) were diagnosed with GDM....

Materials and Methods/Results:

The numbers studied should be included in the results section as well as the numbers excluded for various reasons. This should not be part of the methods section.

Reply: We thank the reviewer for the support. We made the following revisions to the manuscript:

Abstract: “Methods: A retrospective cohort study was performed to assess the prevalence of GDM in women with a triplet pregnancy. GDM was defined through an abnormal oral glucose tolerance test (OGTT). A meta-analysis of GDM prevalence was also carried out. Results: A cohort of 60 women was included in the analysis. Of these, 19 (31.7%) were diagnosed with GDM. […]”

Methods/Results: As recommended, the paragraph on the numbers of studies identified and the exclusion process was moved to the Results Section. However, it was shortened, since the numbers of excluded studies are demonstrated in the new Figure 1.

2) The conclusion is not supported by the results. Firstly, the conclusion states that GDM prevalence is increased in triplet pregnancies but it doesn't define in relation to what is it increased. Is it increased in relation to singleton pregnancies or to twin pregnancies? A baseline prevalence then needs to be given for GDM prevalence in the comparison group (singletons or twins). Secondly, the authors state that 'GDM seems of only minor impact' and yet this minor impact is not described as the outcomes are the same in both GDM and non-GDM groups. Would it be better for the conclusion to state to state that 'GDM did not impact pregnancy outcome'?

Reply: We thank the reviewer for this comment. We revised the abstract’s conclusion: “The rate of GDM seems increased in women with triplets compared to singleton pregnancies. However, GDM did not impact short-term pregnancy outcome.”

Moreover, we also revised the final conclusion as follows: “It can be finally concluded that the rate of GDM seems slightly increased in women with triplets compared to singleton pregnancies. We consider the latter finding sound, since GDM prevalence was evaluated in a well-characterized cohort of women cared for in a tertiary referral centre providing care in line with internationally accepted standards of practice. However, the prevalence was within the range reported for twin pregnancies [8,60-62]. Notably, GDM did not impact short-term pregnancy outcome in the retrospective data set. Whether women with triplets should undergo early screening and whether measures should be taken to reduce associated risks, remains open for future research.”

3) I would suggest removing the reference to fetal weight in line 57. Do the authors mean birth weight or estimated fetal weight on ultrasound? It is obvious that any study of GDM outcomes should include birthweight. I would disagree that they 'focussed on this issue' as it is not analysed more than any other of the secondary outcomes reported. Throughout the paper the authors use the terms 'birthweight', 'neonatal weight' and 'fetal weight' all to apparently mean birthweight. It would be best to use one term consistently.

Reply: We agree with the reviewer. We removed the above mentioned sentence in the Introduction Section. Moreover, we used the term “birth weight” throughout the revised manuscript consistently.

4) Lines 94 and 97 describe baseline characteristics obtained and describe both birthweight (line 94) and neonatal weight (line 97) what is the difference between these? This whole section (line 93-100) needs revision as the units used are quoted for some parameters and not others. This needs to be consistent.

Reply: We apologize for this oversight. As you already assumed in your comment number 3, neonatal and birth weight is the same. The paragraph was revised as follows, since we also included the units for all continuous parameters: “[…]: gestational age at delivery (in completed weeks); maternal age at delivery (years); pre-pregnancy body mass index (BMI, kg/m2); parity; pregnancies after in vitro fertilization (IVF) or ovarian stimulation (i.e. clomiphene citrate, letrozole or recombinant follicle stimulating hormone without IVF); cigarette smoking; pregnancy induced/preexisting hypertension; neonatal birth weight (g); and chorionicity categorized into mono-/dichorionic and trichorionic for the multivariate analysis.”

5) Lines 99-100. This statement seems strange as clinical examination does not rule out all malformations. Do the authors mean that the babies were clinically well? What is the relevance of this to their study? They have not reported other neonatal well-being measures such as Apgar scores, need for resuscitation, cord gas measurements etc.

Reply: We agree that this is of minor or no relevance to the study. We removed the statement and hope that this is fine with the reviewer.

6) It would be helpful to provide a brief description of management of GDM in the Hospital. Are the women seen by both Obstetricians and Diabetologists? How intensively are capillary glucose readings measured? What are the initial interventions? Is it dietary modification and exercise? What is the threshold for initiation of medication? Is Metformin used or is Insulin the first-line medical treatment?

Reply: Thank you for this comment! We added the following information to the Methods Section: “Universal testing by a 75g OGTT at 24–28 weeks of gestation was used to diagnose GDM which is in accordance to the IADPSG recommendations [19,21]. Women were seen by both obstetricians and diabetologists. The first line intervention was intensified lifestyle modification including medical nutrition therapy. All patients were instructed for capillary blood glucose monitoring and informed about glycemic treatment targets. Follow-ups in two weeks' time were scheduled and blood glucose levels were reviewed during each appointment. When blood glucose targets were not achieved (i.e. <95 mg/dl at fasting or <140 mg/dl one hour after each meal), pharmacologic intervention with insulin was started at any time point [22].”

  1. International Association of Diabetes and Pregnancy Study Groups Consensus Panel, "International Association of Diabetes and Pregnancy Study Groups Recommendations on the Diagnosis and Classification of Hyperglycemia in Pregnancy," Diabetes Care, vol. 33, no. 3, pp. 676–682, Mar. 2010.
  2. World Health Organisation, "Diagnostic Criteria and Classification of Hyperglycaemia First Detected in Pregnancy," WHO/NMH/MND/13.2, 2013.
  3. American Diabetes Association, "15. Diabetes Advocacy: Standards of Medical Care in Diabetes-2018," Diabetes Care, vol. 41, no. Suppl 1, pp. S152–S153, Jan. 2018.”

6) Lines 129-32. Why is there a reference to endometriosis? Were there no other variations in the studies except endometriosis prevalence. What relevance does this have to GDM prevalence? This needs to be explained as it is confusing in its current format since there is no prior mention of endometriosis at any stage in the paper.

Reply: We apologize. Probably due to the “COVID-19 situation”, we (the authors) do not function properly. We took parts of the description of the meta-analysis methods from our last meta-analysis on endometriosis prevalence in PCOS – and all of us overlooked it during the proof-reading process. We are sorry. The term “endometriosis” was replaced by “GDM”.

7) Of the 60 pregnancies 14 of the GDM group and 29 of the non-GDM group were the results of IVF or ovarian stimulation. This implies that the other pregnancies were spontaneously conceived. The means that 16/60 triplet pregnancies were spontaneously conceived. This seems to be an abnormally high proportion. Is this correct?

Reply: We are sorry to say that we cannot find these numbers in our manuscript. Table 1 shows that in the group of 19 triplets with GDM, 13 women had conceived with IVF and 1 with ovarian stimulation, whereas in the non-GDM group this had been the case in 25 and 4 patients (out of 41). Thus, two thirds of the women had undergone ART.

8) Table 1: 'Birth weight' is more correct than 'neonatal weight'

Reply: Corrected, thank you.

9) Line 169 - 'notably a wide range of GDM prevalence.. has been reported'. Is this for twin or triplet pregnancies?

Reply: Thank you for this comment. Revised as follows: “Notably, a wide range of the GDM prevalence from about 7% to 20% has been reported for twin pregnancies [8,58-60].”

10) Lines 189-192 - the authors are again commenting on the association between ART and maternal age as they have previously done in lines 50-2. However the maternal age was not different in their GDM and non-GDM groups. Also the mean maternal age in both groups was only 32 years which is relatively young compared to other reports from ART cohorts. The authors should be decisive as to whether or not they feel that maternal age was contributory to GDM prevalence in their cohort. It would be helpful if they compared their triplet cohort to the non-triplet cohort in their hospital in terms of the baseline parameters reported in Table 1.

Reply: We totally agree that a clear statement should be included and revised the paragraph as follows: “Notably, when focusing on our retrospective data set, median maternal age was 32 years, the median BMI was about 23.5kg/m2 (Table 1). Moreover, there were no differences between the GDM- and the non-GDM-groups concerning age and BMI. Thus, we believe that these have not contributed to GDM development.”

Concerning the control group, please see below for details (comment number 12)

11) Lines 206-8. Do the authors really think that the lack of difference in birthweight between the two groups is due to the fact that the GDM women didn't require insulin? This does not seem to be a careful consideration of the factors at work. These are triplet pregnancies with multiple reasons for compromise of fetal growth. Similarly (I assume) these women are being managed and controlled well (see point 6 above). I would see this lack of differences as a reflection on good care being provided for these women. They are also being delivered at 33 weeks gestation, at a time when fetal fat deposition is not at its maximum and a subtle impact of birth weight might not yet be evident. The Discussion could be revised and strengthened by acknowledging these factors.

Reply: We thank the reviewer for this important comment! We took the liberty to use parts of the reviewer’s wording and hope that the reviewer will not get annoyed by this: “Notably, fetal growth can be compromised by various mechanisms in triplet pregnancies [66]. Moreover, the women were being managed and controlled according to our department’s standard procedures (see Methods Section). Thus, the lack of differences in BMI between GDM- and non-GDM-patients could also be seen as a reflection on good care being provided for these women. They were also being delivered at a median gestational age of 33 completed weeks. At this particular time fetal fat deposition is presumably not at its maximum and a subtle impact of birth weight might not yet be evident.”

12) The conclusion needs to be revised. I think this study is important because it presents a value for GDM prevalence in triplet pregnancies in a well-characterised cohort of women cared for in a tertiary referral centre providing care in line with internationally accepted standards of practice. I think this is the most important point of the study. As the authors acknowledge, there are limitations with the data included in the meta-analysis, therefore this is of minor scientific interest. It is necessary to compare the GDM prevalence in triplets to that of either singletons or twins in their centre. The most important finding in this study is that the prevalence of GDM in triplet pregnancies is 32%.

Reply: We thank the reviewer for this input. We were not sure about whether it was reasonable to provide a control group or not. The Medical University of Vienna serves as the reference hospital for high-risk pregnancies in the eastern region of Austria. Thus, women with diverse risk factors are overrepresented in our overall patient population. These risk factors include history of poor pregnancy outcome but also women after bariatric surgery, obesity, internal diseases and so on. Thus, we had to include an and (at least) age- and BMI-matched control group of singleton pregnancies (1:1 matching). The following paragraphs and data were added to the manuscript:

- Methods: “These 60 women with triplet pregnancies were compared to 60 matched women with singleton pregnancies (1:1 matching for age and BMI). The latter were selected from the large population of pregnant women who had undergone first trimester screening from January 2003 to April 2018 and had subsequently delivered at the department.”

- Results: “Nineteen women (31.7%) were diagnosed with an abnormal oGTT. This prevalence exceeded that in age- and BMI-matched singleton pregnancies (11.7%) significantly (p= 0.010). Details on the comparison between these two groups are provided in Table 1.”

- A new Table 1 was added.

- Discussion: “[…], women with triplets are obviously at an increased risk for GDM compared to women with singleton pregnancies, a fact that was also demonstrated by the matched comparison in our patient population (Table 1).”

Discussion: “It can be finally concluded that the rate of GDM seems slightly increased in women with triplets compared to singleton pregnancies. We consider the latter finding sound, since GDM prevalence was evaluated in a well-characterized cohort of women cared for in a tertiary referral centre providing care in line with internationally accepted standards of practice.”

13) Language: It is obvious that this paper is not written by authors who use English as a first language and there are many sentences that read awkwardly. I would recommend review by an English language editor prior to publication. Examples of some areas where language could be improved include:

Abstract:

Line 24 - 'there has been an substantial increase..'

Line 25 - 'twin pregnancies are associated with...'

Line 26 - 'gestational diabetes mellitus (GDM)'

Lines 29 and 136 - diagnosed rather than revealed

Introduction:

Line 40 - 'gestations because of assisted reproductive techniques..'

Lines 42-3 (suggestion) - are clinically important because of their association with both increased maternal and fetal/neonatal morbidity and mortality

Lines 50-2 (suggestion) - Assisted reproductive techniques (ART) are associated with an increase in maternal age compared to spontaneously conceived pregnancies and increased maternal age is in turn independently associated with increased GDM prevalence

Lines 54-5 - 'further data are needed'

Lines 170-4 - these sentences needs to be re-phrased. It is not clear what the authors are trying to say.

Reply: We thank the reviewer for the effort and the suggestions. We revised the sentences as recommended. In addition, the whole manuscript was proof-read by an English language editor.

The last mentioned paragraph (original lines 170-174) was re-written due to the new comparison between triplets and singletons: “It seems obvious that the results of our meta-analyses lie within that range. Thus, one cannot state that women with triplets would carry a higher GDM risk than those with twins. However, women with triplets are obviously at an increased risk for GDM compared to women with singleton pregnancies, a fact that was also demonstrated by the matched comparison in our patient population (Table 1).”

Round 2

Reviewer 1 Report

Thanks to the authors for their answers.

The article is ready to be considered to publish.

Reviewer 2 Report

Thank you for taking my comments on board.  I think the paper is much improved in its current format.